# Therapeutic Modifications without Discontinuation of Atezolizumab Plus Bevacizumab Therapy Are Associated with Favorable Overall Survival and Time to Progression in Patients with Unresectable Hepatocellular Carcinoma

**DOI:** 10.3390/cancers15051568

**Published:** 2023-03-02

**Authors:** Takayuki Tokunaga, Masakuni Tateyama, Yasuteru Kondo, Satoshi Miuma, Shiho Miyase, Kentaro Tanaka, Satoshi Narahara, Hiroki Inada, Sotaro Kurano, Yoko Yoshimaru, Katsuya Nagaoka, Takehisa Watanabe, Hiroko Setoyama, Kotaro Fukubayashi, Motohiko Tanaka, Yasuhito Tanaka

**Affiliations:** 1Department of Gastroenterology and Hepatology, Graduate School of Medical Sciences, Kumamoto University, 1-1-1 Honjo, Chuo-ku, Kumamoto City 860-8556, Kumamoto, Japan; 2Sendai Kousei Hospital, 4-15 Sakamoto, Aoba-ku, Sendai City 980-0873, Miyagi, Japan; 3Department of Gastroenterology and Hepatology, Graduate School of Biomedical Sciences, Nagasaki University, 1-7-1 Sakamoto, Nagasaki City 852-8501, Nagasaki, Japan; 4Kumamoto Shinto General Hospital, 3-2-65 Ooe, Chuo-ku, Kumamoto City 862-8655, Kumamoto, Japan; 5Kumamoto Kenhoku Hospital, 550 Tamana, Tamana City 860-0005, Kumamoto, Japan; 6Public Health and Welfare Bureau, City of Kumamoto, 1-1 Tetori-honcho, Chuo-ku, Kumamoto City 860-8601, Kumamoto, Japan

**Keywords:** hepatocellular carcinoma, atezolizumab plus bevacizumab therapy, therapeutic modifications

## Abstract

**Simple Summary:**

We aimed to evaluate the impact of therapeutic modifications of atezolizumab (Atezo) plus bevacizumab (Bev) therapy (Atezo/Bev) in the case of intolerable adverse events on the prognosis of patients with unresectable hepatocellular carcinoma. Therapeutic modifications included the interruption or discontinuation of both Atezo and Bev, and the reduction, interruption, or discontinuation of Bev alone. Patients with therapeutic modifications other than the discontinuation of both Atezo and Bev had favorable overall survival and time to progression. In contrast, those with the discontinuation of both Atezo and Bev without other therapeutic modifications were associated with unfavorable overall survival and time to progression. Modified albumin–bilirubin grade 2b liver function at the initiation of Atezo/Bev or experience of immune-related adverse events could increase the risk of discontinuation of both Atezo and Bev without other therapeutic modifications. Avoiding the discontinuation of both Atezo and Bev without other therapeutic modifications may be the optimal management of unresectable hepatocellular carcinoma.

**Abstract:**

We retrospectively evaluated the impact of therapeutic modifications of atezolizumab (Atezo) plus bevacizumab (Bev) therapy (Atezo/Bev), including the interruption or discontinuation of both Atezo and Bev, and the reduction or discontinuation of Bev, on the outcome of patients with unresectable hepatocellular carcinoma (uHCC) (median observation period: 9.40 months). One hundred uHCC from five hospitals were included. Therapeutic modifications without discontinuation of both Atezo and Bev (*n* = 46) were associated with favorable overall survival (median not reached; hazard ratio (HR): 0.23) and time to progression (median: 10.00 months; HR: 0.23) with no therapeutic modification defined as the reference. In contrast, the discontinuation of both Atezo and Bev without other therapeutic modifications (*n* = 20) was associated with unfavorable overall survival (median: 9.63 months; HR: 2.72) and time to progression (median: 2.53 months; HR: 2.78). Patients with modified albumin–bilirubin grade 2b liver function (*n* = 43) or immune-related adverse events (irAEs) (*n* = 31) discontinued both Atezo and Bev without other therapeutic modifications more frequently (30.2% and 35.5%, respectively) than those with modified albumin–bilirubin grade 1 (10.2%) and without irAEs (13.0%). Patients with objective response (*n* = 48) experienced irAEs more frequently (*n* = 21) than those without (*n* = 10) (*p* = 0.027). Avoiding the discontinuation of both Atezo and Bev without other therapeutic modifications may be the optimal management of uHCC.

## 1. Introduction

Hepatocellular carcinoma (HCC) is classified into the inflamed class (35%) and the noninflamed class (65%) in the immunogenomic classification [1]. Although chemotherapy using single-agent immune checkpoint inhibitor (ICI) is considered not effective in the noninflamed HCC, a combination of ICI and anti-vascular endothelial growth factor (VEGF) antibodies can be effective in such an immunosuppressive tumor microenvironment [2,3].

In a phase 3 trial of first-line chemotherapy in patients with unresectable HCC (uHCC) (IMbrave150 trial), Atezolizumab (Atezo) plus Bevacizumab (Bev) therapy (Atezo/Bev) was associated with superior overall survival (OS), progression-free survival, and objective response (OR) rate compared with sorafenib therapy [4,5]. Atezo is an ICI that targets the programmed death ligand 1 (PD-L1) and blocks immune evasion by tumors. Bev is a monoclonal antibody that targets VEGF and blocks VEGF-mediated immunosuppression and angiogenesis in tumors. Immune-related adverse events (irAEs) caused by Atezo or adverse events (AEs) induced by Bev can occur during Atezo/Bev. In the IMbrave150 trial, the interruption of Atezo or Bev and discontinuation of Atezo or Bev were carried out in response to the occurrence of intolerable AEs [4,5]. In a phase 2 study of Bev monotherapy in patients with uHCC, a reduction in the Bev dosage was also permitted [6]. However, the impacts of therapeutic modifications and AEs, including irAEs, on the outcome of Atezo/Bev remain to be clarified.

In this study, the impact of therapeutic modifications on the OS, time to progression (TTP), and achievement of OR in patients with uHCC receiving Atezo/Bev was assessed. The aim was to determine the optimal management in real-world practice.

## 2. Patients and Methods

### 2.1. Patients

This multicenter, retrospective study was conducted in Japan, and included the Kumamoto University Hospital, the Sendai Kousei Hospital, the Nagasaki University Hospital, the Kumamoto Shinto General Hospital and the Kumamoto Kenhoku Hospital. Consecutive patients with uHCC who initiated Atezo/Bev as first- or later-line therapy from December 2018 to April 2022 were enrolled. Data were censored as of the end of September 2022.

The inclusion criteria were: (a) radiologically typical or pathologically confirmed HCC; (b) Child–Pugh (CP) score of 5–9; and (c) Eastern Cooperative Oncology Group performance status (ECOG-PS) of 0–1. The exclusion criteria were: (a) renal dysfunction (serum creatinine: >1.5-fold higher than the upper limit of the normal range); (b) chronic heart failure; (c) inadequate hematological function (platelet count: <50 × 10^9^/L; hemoglobin: <8.0 g/dL or white blood cell count: <2000/mm^3^); and (d) alanine aminotransferase or aspartate aminotransferase concentration >5-fold higher than the upper limit of the normal range. All the patients in this study were assessed with esophagogastroduodenoscopy before the initiation of Atezo/Bev. The initiation of Atezo/Bev preceded the treatment of high-risk varix in the case where progression of HCC was predicted during the treatment of varix.

Currently, the indication for Atezo/Bev in patients with CP class B is debatable [7]. Nevertheless, the American Association for the Study of Liver Diseases recommends the use of systemic therapy for well-selected HCC patients with CP class B [8]. In line with this recommendation, we included patients with ECOG-PS 0–1 and CP class B who did not meet the exclusion criteria as “well-selected HCC patients with CP class B liver function”, as we previously reported [9].

### 2.2. Radiological Diagnosis of HCC

Typical HCC was defined as follows: hypervascularity in late arterial phase and wash-out on portal venous and/or delayed phases, detected by multiphasic-computed tomography or dynamic contrast-enhanced magnetic resonance imaging [10,11,12,13].

### 2.3. Practice of Atezo/Bev

In each cycle of Atezo/Bev, patients received both 1200 mg of Atezo and 15 mg/kg of Bev intravenously every 3 weeks (standard dose). A reduced starting dose of Bev was considered for patients with a history of AEs induced by prior VEGF-targeted therapies, such as proteinuria or hemorrhage.

Therapeutic modifications of Atezo/Bev, including a reduction in Bev dose, interruption of Atezo, Bev, and both Atezo and Bev, as well as the discontinuation or interruption of Atezo, Bev, and both Atezo and Bev, were permitted in case of intolerable AEs and the deterioration of liver function or ECOG-PS. A reduction in Bev dose was defined as a reduction in the most recently administered dosage. Interruption was defined as a temporary discontinuation of Atezo, Bev, or both Atezo and Bev. After improvement of AEs, treatment with Atezo at the same dose or Bev at the same or a reduced dose was re-initiated. Re-initiation of treatment with Bev at a lower dose than that at the time of interruption was also classified as dose reduction. Atezo/Bev was continued until radiological or clinical progression.

### 2.4. Evaluation of Atezo/Bev

The best response to Atezo/Bev was evaluated by the experienced radiologist and hepatologist according to modified Response Evaluation Criteria in Solid Tumors (mRECIST) as complete response (CR), partial response (PR), stable disease (SD), and progressive disease (PD). OR was determined as CR plus PR [10]. Radiological assessment was performed at baseline, 4–8 weeks after the initiation of Atezo/Bev, and every 2–3 months thereafter.

OS was defined as the interval from the date of Atezo/Bev initiation to that of death or final observation. TTP was defined as the interval from the date of Atezo/Bev initiation to that of progression or final radiological assessment. Time to confirmation of best response (TTBR) was calculated as the interval from the date of Atezo/Bev initiation to that of the first achievement of the best response.

Therapeutic modifications of Atezo/Bev performed during TTP and until TTBR were classified as follows: (a) discontinuation of both Atezo and Bev without other therapeutic modifications (discontinuation of both Atezo and Bev alone); (b) discontinuation of both Atezo and Bev with other therapeutic modifications; (c) therapeutic modifications other than discontinuation of both Atezo and Bev; and (d) no therapeutic modification.

AEs were evaluated based on the Common Terminology Criteria for Adverse Events version 5.0. In line with the AEs of special interest evaluated in the IMbrave150 trial, each AE was classified into irAE or AE other than irAE. Furthermore, the timing of each therapeutic modification or AE experience was classified as occurring before or after TTBR to evaluate the impact on best response.

### 2.5. Relative Dose Intensity (RDI) of Atezo and Bev

The RDI of Atezo and Bev was evaluated until progression or final radiological assessment by calculating the ratio of the administered dose to the standard dose. The standard dose of Atezo and Bev was calculated based on the number of standard cycles of Atezo/Bev administered during TTP. The number of standard cycles of Atezo/Bev was calculated by dividing the TTP (days) by 21 days and rounding up the first decimal place to the nearest integer.

### 2.6. Study Assessment

Firstly, factors associated with OS and TTP, including characteristics at the initiation of Atezo/Bev, best response, experience of AEs during TTP, and experience of therapeutic modifications during TTP, were analyzed in all the patients as well as in the patients with OR and SD at best response. Regarding time-dependent factors of best response and experience of therapeutic modifications, landmark analyses of OS and TTP at 2, 4, and 6 months after the initiation were performed. Patients who died and progressed before the landmark point were excluded from analysis of OS and TTP, respectively.

Secondly, factors associated with the achievement of OR, including characteristics at the initiation of Atezo/Bev, therapeutic modifications performed until TTBR, and AEs experienced until TTBR, were evaluated. Thirdly, factors associated with the discontinuation of both Atezo and Bev because of AEs without other therapeutic modifications, including characteristics at the initiation of Atezo/Bev and AEs experienced during TTP, were analyzed. Finally, the relationships between best response and RDI, therapeutic modification, and AEs were evaluated during TTP, as well as until and after TTBR.

### 2.7. Statistical Analysis

OS and TTP were analyzed using the Kaplan–Meier method. Cox regression analysis was performed to analyze the factors associated with OS and TTP. Binomial logistic regression analysis was used to analyze the factors associated with the achievement of OR and discontinuation of both Atezo and Bev because of AEs without other therapeutic modifications.

Regarding factors at the initiation of treatment, we adopted 50% as the cut-off value for the intrahepatic tumor volume. This was in line with the exclusion criteria applied to a phase 3 trial of lenvatinib in patients with uHCC [14]. We also adopted 6 cm and six nodules as the cut-off values for tumor size and tumor number, respectively, according to previous reports [15]. Continuous variables were categorized into two groups; the median was used as the cut-off value for age, while 400 ng/mL was used as the cutoff value for the levels of alpha-fetoprotein. Liver function was assessed based on the CP class and modified albumin–bilirubin (mALBI) grade, in accordance with a previous report [16].

Two-tailed probabilities were used, with *p*-values ≤ 0.05 denoting statistically significant differences. All factors that exhibited statistical significance in the univariate analysis were subsequently included in the multivariate analysis. All statistical analyses were conducted using the SPSS software (SPSS Statistics version 27; IBM Corp., Armonk, NY, USA).

## 3. Results

### 3.1. Patient Characteristcs

A total of 100 patients were enrolled in this study (Figure 1). A total of 84 patients (84.0%) had CP class A liver function, while 29, 27, 43, and 1 patients were categorized into mALBI grades 1, 2a, 2b, and 3, respectively. A total of 15 patients had macrovascular invasion and 3 patients showed intrahepatic tumor volume ≥50%. A total of 58 and 42 patients received Atezo/Bev as first- and second- or later-line chemotherapy, respectively. Thirty-nine patients had esophageal varices. Seven of them had varices with a high risk of bleeding, but they started Atezo/Bev without receiving preventative endoscopic therapy because progression of HCC was predicted during the treatment of the varices. Seven patients had gastric varices, all of which had a low risk of bleeding (Table 1). The median observation period was 9.40 months (range: 1.11–22.22 months).

### 3.2. Outcome of Atezo/Bev

The median OS was 21.90 months (95% confidence interval [CI]: 16.86–26.93 months) and the median TTP was 5.79 months (95% CI: 3.69–7.88 months) (Figure 2). In terms of best response, 7, 41, 38, and 14 patients had CR, PR, SD, and PD, respectively. The TTBR for OR, SD, and PD was 1.61 months (95% CI: 1.20–2.02 months), 1.25 months (95% CI: 1.17–1.33 months), and 0.95 months (95% CI: 0.59–1.32 months), respectively (Figure 3). A total of 66 patients experienced progression. Among them, 58 patients had radiological progression without clinical progression, 2 patients had both radiological and clinical progression, and 6 patients had clinical progression without radiological progression.

### 3.3. Therapeutic Modifications, RDI, and AEs of Atezo/Bev

Ninety-five patients (95.0%) initiated Atezo/Bev at the standard dose, while five patients initiated treatment with Bev at a reduced dose. Those five patients included one (7.5 mg/kg) with a history of tracheal hemorrhage of grade 1 that occurred during previous lenvatinib therapy, one (12 mg/kg) with proteinuria caused by chronic kidney disease, and three (7.5 mg/kg) with ascites (Table 1). The median RDI of Atezo and Bev were 0.92 (range: 0.11–1.00) and 0.76 (range: 0.11–1.00), respectively.

Interruption of both Atezo and Bev was carried out in 35 patients (Table 2). The median time to first interruption was 12.56 months (95% CI: 6.33–18.78 months). The cumulative first interruption rates at 6 weeks and 6 months were 16.9% and 33.1%, respectively (Figure 4a). The most frequent reason for interruption was implantation of a central venous access port (*n* = 7) for the continuation of Atezo/Bev. One patient, who had high-risk esophageal varix at the screening, experienced the rupture, resulting in interruption of both Atezo and Bev (Table 3). Twenty-nine patients restarted treatment with both agents, and the median time to re-initiation was 17 days (range: 5–104 days). Of those, ten patients restarted Atezo alone, and the median time to re-initiation was 30.5 days (range: 7–140 days). Moreover, three patients restarted Bev alone, and the median time to re-initiation was 77 days (range: 27–226 days).

Discontinuation of both Atezo and Bev because of AEs was carried out in 25 patients (Table 2). The median time to first discontinuation was 19.00 months (95% CI: 13.50–24.50 months), and the cumulative discontinuation rates at 6 weeks and 6 months were 11.4% and 23.4%, respectively (Figure 4b). Ascites (*n* = 3) was the most frequent reason for discontinuation of both Atezo and Bev (Table 3).

Reduction in Bev dose was performed in 12 patients (Table 2). The median time to first reduction was not reached, and the cumulative dose reduction rates at 6 weeks and 6 months were 1.1% and 12.2%, respectively (Figure 4c). Proteinuria (*n* = 4) was the most frequent reason for the reduction in Bev dose (Table 3).

Interruption of Bev was carried out in 17 patients (Table 2). The median time to first interruption was not reached, and the cumulative interruption rates at 6 weeks and 6 months were 1.0% and 12.9%, respectively (Figure 4d). The median time to re-initiation was 42 days (range: 21–123 days). Proteinuria (*n* = 8) was the most frequent reason for interruption of Bev (Table 3). Interruption of Atezo alone was not carried out in any of the patients.

Discontinuation of Bev was performed in eight patients (Table 2). The median time to first discontinuation was not reached, and the cumulative dose reduction rates at 6 weeks and 6 months were 2.1% and 6.2%, respectively (Figure 4e). Proteinuria (*n* = 6) was the most frequent reason for the discontinuation of Bev (Table 3). Discontinuation of Atezo alone was not performed in any of the patients.

### 3.4. Factors Associated with OS and Factors Associated with TTP

In all the enrolled patients, achievement of OR was a favorable factor of OS (hazard ratio (HR): 0.070; 95% CI: 0.015–0.32; *p* < 0.001) and TTP (HR: 0.013; 95% CI: 0.0050–0.037; *p* < 0.001), with PD used as the reference. Experience of therapeutic modifications other than discontinuation of both Atezo and Bev (Table 4) was also associated with favorable OS (HR: 0.23; 95% CI: 0.075–0.68; *p* = 0.0080) and favorable TTP (HR: 0.27; 95% CI: 0.057–1.28; *p* = 0.0040), with no therapeutic modification used as the reference. Nevertheless, experience of discontinuation of both Atezo and Bev alone was associated with unfavorable OS (HR: 2.72; 95% CI: 1.05–7.07; *p* = 0.040) and unfavorable TTP (HR: 2.72; 95% CI: 1.05–7.07; *p* = 0.040), with no therapeutic modification used as the reference. Notably, discontinuation of both Atezo and Bev with other therapeutic modifications (Table 4) was not associated with OS or TTP (Table 5).

In the patients who progressed on Atezo/Bev, achievement of OR (HR: 0.11; 95% CI: 0.024–0.50; *p* = 0.0040) and experience of therapeutic modifications other than discontinuation of both Atezo and Bev (HR: 0.24; 95% CI: 0.079–0.70; *p* = 0.0090) were favorable factors of OS with PD and no therapeutic modification used as the reference, respectively, while experience of subsequent therapy (*n* = 31) was not an independent factor of OS (Appendix A).

Furthermore, in the patients with OR and SD, achievement of OR (HR: 0.085; 95% CI: 0.018–0.40; *p* = 0.0020) was associated with favorable OS (HR: 4.93; 95% CI: 1.37–17.74; *p* = 0.015) and favorable TTP (HR: 0.26; 95% CI: 0.14–0.47; *p* < 0.001), while experience of discontinuation of both Atezo and Bev alone was associated with unfavorable OS (HR: 4.93; 95% CI: 1.37–17.74; *p* = 0.015) and unfavorable TTP (HR: 3.59; 95% CI: 1.57–8.23; *p* = 0.0030), with no therapeutic modification used as the reference; meanwhile, therapeutic modifications other than the discontinuation of both Atezo and Bev as well as the discontinuation of both Atezo and Bev with other therapeutic modifications were not associated with OS or TTP (Appendix A).

### 3.5. Land Mark Analyses of OS and TTP Regarding Best Response and Therapeutic Modifications

Regarding best response, the patients with OR had favorable OS compared with SD at 2, 4, and 6 months (*p* = 0.039, 0.001 and <0.001) or PD at 2, 4, and 6 months (*p* < 0.001, <0.001 and <0.001) (Appendix A). The patients with OR had favorable TTP compared with PD at 2 months (*p* < 0.001), while those with OR had similar TTP compared with SD at 2, 4, and 6 months (Appendix A).

Concerning therapeutic modifications, the patients with therapeutic modifications other than the discontinuation of both Atezo and Bev had favorable OS at 2 months (*p* < 0.001), similar OS at 4 and 6 months, and similar TTP at 2, 4, and 6 months, compared with those with no therapeutic modification (Appendix A). The patients with discontinuation of both Atezo and Bev alone had unfavorable OS at 2 months (*p* < 0.001), and unfavorable TTP at 2 and 4 months (*p* = 0.014 and 0.02) (Appendix A).

### 3.6. Factors Associated with the Achievement of OR and Discontinuation of Both Atezo and Bev

Regarding the achievement of OR, the administration of Atezo/Bev in the first-line setting (odds ratio: 3.89; 95% CI: 1.59–9.52; *p* = 0.0030) was a favorable factor. Vascular invasion (odds ratio: 0.24; 95% CI: 0.058–0.99; *p* = 0.048) and a maximum tumor size >6 cm (odds ratio: 0.32; 95% CI: 0.11–0.91; *p* = 0.033) were negatively associated with the achievement of OR. Experience of therapeutic modifications until TTBR was not associated with the achievement of OR (Table 5).

Concerning the discontinuation of both Atezo and Bev alone due to AEs, the occurrence of irAEs of any grade was an unfavorable factor (odds ratio: 3.57; 95% CI: 1.26–10.13; *p* = 0.017). Patients with mALBI grade 2 liver function discontinued both Atezo and Bev without other therapeutic modifications more frequently than those with mALBI grade 1 liver function (30.2% vs. 10.3%, respectively) (Table 5).

### 3.7. Relationships between Best Response and RDI, Therapeutic Modification, and AEs during TTP, until TTBR, and after TTBR

Regarding the relationships between best response and RDI, patients with OR had lower RDI of Atezo during TTP than those with SD and PD (0.88, 0.96, and 1.00, respectively; *p* = 0.069), and after TTBR than those with SD (0.84 and 0.80, respectively; *p* = 0.062). Moreover, patients with OR had lower RDI of Bev during TTP than those with SD and PD (0.70, 0.76, and 1.00, respectively; *p* = 0.016) (Table 6).

With regard to the relationships between best response and therapeutic modifications, patients with OR experienced an interruption of both Atezo and Bev more frequently than those with SD and PD during TTP (65.7%, 28.6%, and 5.7%, respectively; *p* = 0.024) and after TTBR (77.8%, 21.2%, and 0.0%, respectively; *p* < 0.001). An interruption of Bev occurred more frequently in patients with OR than those with SD and PD during TTP (70.6%, 29.4%, and 0.0%, respectively; *p* = 0.066) (Table 2).

Concerning the relationships between AEs and therapeutic modifications, patients with OR experienced AEs of grade ≥ 3 more frequently than those with SD and PD during TTP (64.5%, 32.3%, and 3.2%, respectively; *p* = 0.036). Moreover, patients with OR experienced irAEs of any grade more frequently than those with SD and PD during TTP (67.7%, 25.8%, and 6.5%, respectively; *p* = 0.027) and after TTBR (80.0%, 20.0%, and 0.0%, respectively; *p* = 0.021) (Table 2).

## 4. Discussion

In this study, we evaluated the impact of therapeutic modifications of Atezo/Bev on the OS, TTP, and the achievement of OR in patients with uHCC over an adequate observation period (median: 9.40 months). Firstly, we found that patients who experienced therapeutic modifications other than discontinuation of both Atezo and Bev were associated with favorable OS and TTP. Secondly, those who experienced discontinuation of both Atezo and Bev because of intolerable AEs without other therapeutic modifications had unfavorable TTP and OS. Thirdly, therapeutic modifications performed until TTBR did not contribute to the achievement of OR. Therefore, therapeutic modifications other than the discontinuation of both Atezo and Bev in case of intolerable AEs may not impair the outcome of patients with uHCC receiving Atezo/Bev. Additionally, the discontinuation of both Atezo and Bev with other therapeutic modifications did not impair the treatment outcome. Collectively, the present evidence indicates that therapeutic modifications other than the discontinuation of both Atezo and Bev may mitigate the toxicity of this combination therapy [17].

The discontinuation of both Atezo and Bev was carried out more frequently in this study (25.2%) than the IMbrave150 trial (10.0%) [5]. This may be because patients in the present study had worse liver function than those included in the IMbrave150 trial. In fact, those with mALBI grade 2b liver function at treatment initiation discontinued both Atezo and Bev without other therapeutic modifications more frequently than those with mALBI grade 1 liver function. This is consistent with previously reported evidence [18]. It was also previously reported that mALBI grade 2b liver function contributed to an early interruption of Bev within the first 9 weeks of treatment (26.3%); moreover, those experiencing an early interruption of Bev were linked to an unfavorable OS, unfavorable progression-free survival, and lower OR rate [19]. In this study, patients who experienced therapeutic modifications other than a discontinuation of both Atezo and Bev, including an interruption and discontinuation of Bev, had favorable OS and TTP. This discrepancy may be attributed to differences in therapeutic modifications and the observation period. In fact, the interruption and discontinuation of Bev were carried out within the first 9 weeks (1.0% and 3.2%, respectively) less frequently in this study versus a previous investigation [19].

In this study, patients who achieved OR (assessed using mRECIST) were associated with favorable TTP and OS. This is in accordance with evidence reported in a systemic review of randomized clinical trials of molecular targeted therapy in patients with uHCC, including the IMbrave150 trial [20] and its updated data [21]. Therefore, in patients with OR, it may be possible to allocate some time for the management of intolerable AEs, while maintaining therapeutic response. In this study, patients with OR experienced an interruption of both Atezo and Bev, as well as AEs of grade ≥3 more frequently than those without OR. Moreover, those with OR had lower RDI of Atezo and Bev versus those without OR. Furthermore, patients who experienced irAEs discontinued both Atezo and Bev without other therapeutic modifications more frequently than those who did not experience such events. Hence, it is conceivable that the prediction of irAEs prior to the initiation of Atezo/Bev is crucial for treatment success. We found that patients with OR also experienced irAEs of any grade during TTP, especially after TTBR, more frequently than those without. Therefore, the occurrence of irAEs might be expected after the confirmation of OR using mRECIST in patients with uHCC receiving Atezo/Bev. It remains unclear whether the circumstances under which irAEs occur contribute to the achievement of OR. However, a post hoc analysis of clinical trials of nivolumab therapy in patients with advanced melanoma revealed a higher rate of OR in those with irAEs than those without [22].

In this study, the median TTBR for OR based on mRECIST was shorter than that recorded in the IMbrave150 trial (1.61 vs. 2.8 months, respectively). This difference might be associated with the smaller size of liver lesions in this study compared with the IMbrave150 trial. In that trial, the maximum tumor size was smaller in patients with CR versus those without [20]. Similarly, in this study, patients with a maximum tumor size >6 cm achieved OR less frequently than those with a maximum tumor size ≤6 cm. Our data suggested that the OR rate after second- or later-line therapy could be lower than that achieved in the first-line setting. The resistance of tumors to prior anti-VEGF therapy may partly explain this difference, as previously reported [23]. This finding may be important when another combination of cancer immunotherapy and anti-VEGF therapy is approved as first-line therapy for patients with uHCC in the future.

Although neither esophageal varix nor gastric varix were associated with OS, TTP, achievement of OR, and discontinuation of both Atezo and Bev without other therapeutic modifications in this study, receiving therapy to prevent the bleeding of varix before the initiation of Atezo/Bev would be desirable in line with the IMbrave 150 trial [4], if high-risk varix is detected at the screening and the patients have time to spare to wait for the initiation of Atezo/Bev. The management of the patients who had no time to spare to receive the preventative therapy for varix before Atezo/Bev should be solved in the future [24,25].

The limitations of this study should be acknowledged. Firstly, this was a retrospective study; hence, biases from unobserved differences must be considered. Secondly, in this study, we examined the impact of therapeutic modifications during TTP on OS and TTP, as well as that of therapeutic modifications until TTBR on the achievement of OR. However, we could not assess the impact of therapeutic modifications during Atezo/Bev on OS. Finally, the sample size of this study is small. Further prospective studies with a larger number of patients are warranted to confirm the present results.

## 5. Conclusions

The present findings reveal the impact of therapeutic modifications in case of intolerable AEs on the prognosis of patients with uHCC receiving Atezo/Bev. Favorable prognosis was observed for patients in whom therapeutic modifications other than the discontinuation of both Atezo and Bev were performed. In contrast, unfavorable prognosis was noted for patients in whom both Atezo and Bev were discontinued without other therapeutic modifications due to the occurrence of AEs. Therefore, avoiding the discontinuation of both Atezo and Bev without other therapeutic modifications may be the optimal management of uHCC.

Modified ALBI grade 2b liver function at the initiation of Atezo/Bev or experience of irAEs during this treatment could increase the risk of discontinuation of both Atezo and Bev without other therapeutic modifications. Hence, the close monitoring of patients with mALBI grade 2b liver function at the time of treatment initiation is warranted. Importantly, the occurrence of irAEs might be expected after the achievement of OR.

## Figures and Tables

**Figure 1 cancers-15-01568-f001:**
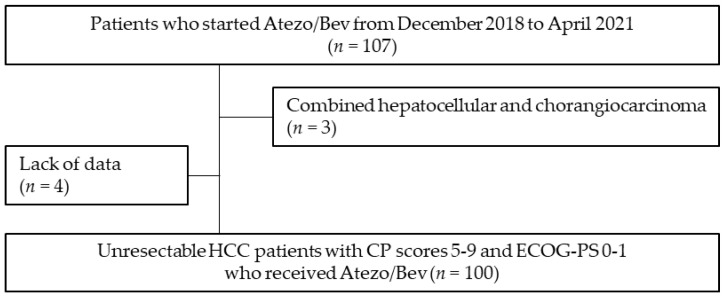
Flowchart of the study design. Abbreviations: Atezo/Bev, atezolizumab plus bevacizumab therapy; CP, Child–Pugh; ECOG-PS, Eastern Cooperative Oncology Group performance status; HCC, hepatocellular carcinoma.

**Figure 2 cancers-15-01568-f002:**
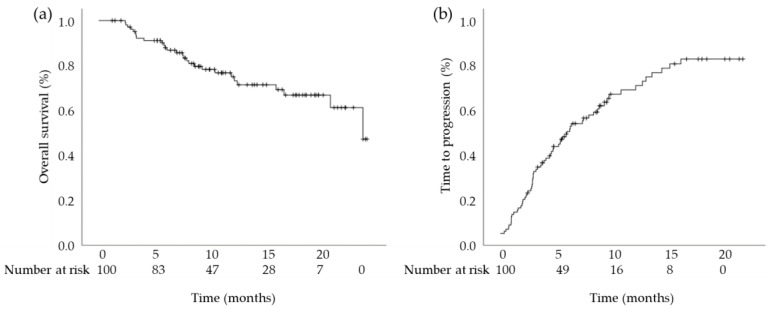
Kaplan–Meier estimates of cumulative (**a**) overall survival and (**b**) time to progression.

**Figure 3 cancers-15-01568-f003:**
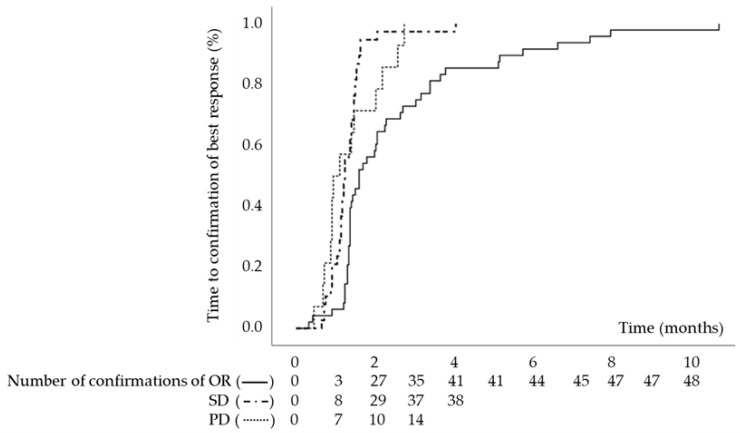
Kaplan–Meier estimate of cumulative time to the confirmation of best response. Abbreviations: OR, objective response; PD, progressive disease; SD, stable disease.

**Figure 4 cancers-15-01568-f004:**
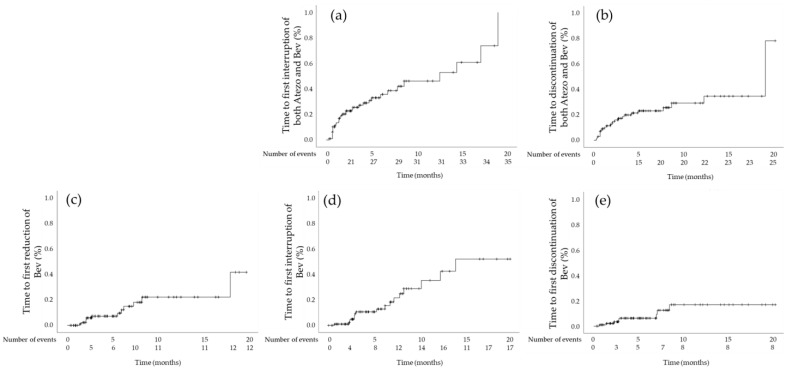
Kaplan–Meier estimates of cumulative (**a**) time to first interruption of both Atezo and Bev; (**b**) time to discontinuation of both Atezo and Bev; (**c**) time to first reduction in Bev dose; (**d**) time to first interruption of Bev; and (**e**) time to discontinuation of Bev. Abbreviations: Atezo, atezolizumab; Bev, bevacizumab.

**Table 1 cancers-15-01568-t001:** Baseline characteristics of patients (*n* = 100).

Characteristics	Number of Patients (%)
Age (≥70 years)	59 (59.0)
Sex (Male)	84 (84.0)
HBs antigen (Positive)	20 (20.0)
HCV antibody (Positive)	30 (30.0)
Alcoholic liver disease (Present)	35 (35.0)
Esophageal varix (Present)	39 (39.0)
Gastric varix (Present)	7 (7.0)
Autoimmune diseases (Present)	0 (0.0)
History of previous curative therapy (Present)	54 (54.0)
History of previous TACE (Present)	65 (65.0)
Child–Pugh class (A/B)	84 (84.0)/16 (16.0)
Modified ALBI grade (1/2a/2b/3)	29 (29.0)/27 (27.0)/43 (43.0)/1 (1.0)
ECOG-PS (0/1)	86 (86.0)/14 (14.0)
BCLC stage (A/B/C)	3 (3.0)/44 (44.0)/53 (53.0)
Maximum tumor size (>6 cm)	74 (74.0)
Tumor number (>6)	47 (47.0)
Intrahepatic tumor volume (≥50%)	3 (3.0)
Macrovascular invasion (Present)	15 (15.0)
Extrahepatic spread (Present)	41 (41.0)
AFP concentration (≥400 ng/mL)	28 (28.0)
Initial dose of atezolizumab (Standard dose)	100 (100.0)
Initial dose of bevacizumab (Reduced dose)	5 (5.0)
Number of chemotherapy lines	
First-line	58 (58.0)
Second- or later-line	42 (42.0)
History of sorafenib therapy (Present)	14 (14.0)
History of regorafenib therapy (Present)	4 (4.0)
History of lenvatinib therapy (Present)	36 (36.0)
History of ramucirumab therapy (Present)	2 (2.0)

Variables are expressed as number (%). Abbreviations: AFP, alpha-fetoprotein; ALBI, albumin–bilirubin; BCLC, Barcelona Clinic Liver Cancer; ECOG-PS, Eastern Cooperative Oncology Group performance status; HCV, hepatitis C virus; HBs, hepatitis B surface; TACE, transarterial chemoembolization.

**Table 2 cancers-15-01568-t002:** Relationships between best response, therapeutic modification, and AEs experienced during TTP, until, and after TTBR.

Therapeutic Modifications and AEs	Best Response, *n* (%)			*p*-Value	RDI		Administered Cycles		Number of Events
	OR (*n* = 48)	SD (*n* = 38)	PD (*n* = 14)		Atezo	Bev	Atezo	Bev	
Interruption of both Atezo and Bev (*n* = 35)	23 (65.7%)	10 (28.6%)	2 (5.7%)	**0.024**	0.75 (0.12–0.92)	0.67 (0.13–0.90)	9 (1–20)	6 (1–18)	1 (1–10)
Until TTBR (*n* = 17)	9 (52.9%)	9 (52.9%)	2 (11.8%)	0.90	0.75 (0.33–0.88)	0.62 (0.15–0.86)	9 (1–20)	6 (1–18)	1 (1–10)
After TTBR (*n* = 27)	21 (77.8%)	6 (21.2%)	0 (0.0%)	**<0.001**	0.79 (0.12–0.92)	0.67 (0.13–0.90)	9 (2–20)	8.5 (1–18)	1 (1–9)
Discontinuation of both Atezo and Bev (*n* = 25)	11 (44.0%)	11 (44.0%)	3 (12.0%)	0.77	0.67 (0.11–1.00)	0.50 (0.11–1.00)	2 (1–20)	2 (1–18)	1 (1)
Until TTBR (*n* = 12)	5 (41.7%)	4 (33.3%)	3 (25.0%)	0.40	0.43 (0.11–1.00)	0.41 (0.11–1.00)	2 (1–12)	2 (1–12)	1 (1)
After TTBR (*n* = 13)	6 (46.2%)	7 (53.8%)	0 (0.0%)		0.69 (0.44–1.00)	0.69 (0.44–1.00)	5 (2–20)	5 (2–18)	1 (1)
Reduction in Bev dose (*n* = 12)	8 (66.7%)	4 (33.3%)	0 (0.0%)	0.23	0.87 (0.12–1.00)	0.47 (0.15–0.91)	9.5 (3–21)	6 (3–20)	1 (1–2)
Until TTBR (*n* = 2)	2 (100.0%)	0 (0.0%)	0 (0.0%)	0.33	0.36 (0.12–0.61)	0.36 (0.24–0.48)	8.5 (3–14)	12 (6–18)	1 (1)
After TTBR (*n* = 11)	7 (63.6%)	4 (36.4%)	0 (0.0%)	0.31	0.92 (0.12–1.00)	0.48 (0.15–0.91)	9 (3–21)	6 (3–20)	1 (1–2)
Interruption of Bev (*n* = 17)	12 (70.6%)	5 (29.4%)	0 (0.0%)	0.066	0.89 (0.46–1.00)	0.67 (0.24–0.85)	12 (5–21)	10 (4–20)	1 (1–6)
Until TTBR (*n* = 4)	3 (75.0%)	1 (25.0%)	0 (0.0%)	0.50	0.80 (0.46–1.00)	0.39 (0.24–0.70)	13.5 (5–20)	9 (4–14)	1.5 (1–4)
After TTBR (*n* = 14)	10 (71.4%)	4 (28.6%)	0 (0.0%)	0.10	0.88 (0.61–1.00)	0.71 (0.24–0.85)	11.5 (7–21)	9 (4–20)	1 (1–6)
Discontinuation of Bev (*n* = 8)	4 (50.0%)	3 (37.5%)	1 (12.5%)	0.99	1.00 (0.55–1.00)	0.41 (0.13–0.83)	11.5 (4–17)	4 (1–12)	1 (1)
Until TTBR (*n* = 3)	2 (66.7%)	0 (0.0%)	1 (33.3%)	0.47	1.00 (0.63–1.00)	0.33 (0.13–0.50)	5 (4–12)	2 (1–4)	1 (1)
After TTBR (*n* = 5)	2 (40.0%)	3 (60.0%)	0 (0.0%)		1.00 (0.55–1.00)	0.48 (0.18–0.83)	12 (8–17)	6 (3–12)	1 (1)
AEs of grade ≥3 (*n* = 31)	20 (64.5%)	10 (32.3%)	1 (3.2%)	**0.036**	0.89 (0.12–1.00)	0.71 (0.13–1.00)	9 (1–27)	6 (1–27)	1 (1–2)
Until TTBR (*n* = 15)	10 (66.7%)	4 (26.7%)	1 (6.7%)	0.28	0.86 (0.12–1.00)	0.50 (0.13–0.85)	5 (1–27)	6 (1–27)	1 (1)
After TTBR (*n* = 18)	11 (61.1%)	7 (38.9%)	0 (0.0%)	0.15	0.89 (0.33–1.00)	0.70 (0.13–1.00)	9.5 (2–17)	7.0 (1–14)	1 (1)
IrAEs of any grade (*n* = 31)	21 (67.7%)	8 (25.8%)	2 (6.5%)	**0.027**	0.75 (0.12–1.00)	0.69 (0.13–1.00)	7 (1–20)	6 (1–18)	1 (1–2)
Until TTBR (*n* = 17)	10 (58.8%)	5 (29.4%)	2 (11.8%)	0.62	0.71 (0.12–1.00)	0.75 (0.13–1.00)	5 (1–13)	5 (1–18)	1 (1)
After TTBR (*n* = 15)	12 (80.0%)	3 (20.0%)	0 (0.0%)	**0.021**	0.75 (0.33–1.00)	0.75 (0.15–1.00)	9 (1–20)	9 (1–18)	1 (1)
AEs other than irAEs of grade ≥ 3 (*n* = 27)	17 (63.0%)	9 (33.3%)	1 (3.7%)	0.094	0.90 (0.46–1.00)	0.76 (0.13–1.00)	9 (1–27)	7 (1–27)	1 (1–2)
Until TTBR (*n* = 12)	8 (66.7%)	3 (25.0%)	1 (8.3%)	0.39	0.91 (0.50–1.00)	0.81 (0.13–1.00)	5.5 (1–27)	5.5 (1–27)	1 (1)
After TTBR (*n* = 17)	10 (58.8%)	7 (41.2%)	0 (0.0%)	0.18	0.89 (0.46–1.00)	0.71 (0.13–1.00)	9 (1–27)	8 (1–27)	1 (1)

Abbreviations: AEs; adverse events; Atezo, atezolizumab; Bev, bevacizumab; irAEs, immune-related adverse events; OR, objective response; PD, progressive disease; RDI, relative dose intensity; SD, stable disease; TTBR, time to confirmation of best response; TTP, time to progression. Values for relative dose intensity, number of administered cycles, and number of events are presented as the median (range). Bold font indicates statistically significant *p*-values.

**Table 3 cancers-15-01568-t003:** AEs and events which caused therapeutic modifications of Atezo/Bev during time to progression (*n* = 100).

	Grade of AEs		Therapeutic Modification of Both Atezo and Bev		Therapeutic Modification of Bev		
	Any	≥3	Interruption	Discontinuation	Reduction	Interruption	Discontinuation
Immune-related AEs							
Rash	12	0	0	1	0	0	0
Hypothyroidism	6	0	0	0	0	0	0
Hyperthyroidism	4	0	0	0	0	0	0
Infusion-related reaction	4	0	0	0	0	0	0
Pneumonitis	3	1	1	2	0	0	0
Hepatitis	3	2	1	2	0	0	0
Adrenal insufficiency	2	0	0	1	0	0	0
Hypopituitarism	2	1	0	1	0	0	0
Colitis	1	1	2	0	0	0	0
Myositis	1	0	0	1	0	0	0
Sjögren syndrome	1	0	1	0	0	0	0
AEs other than immune-related AEs							
Increase in AST	48	2	0	0	0	0	0
Hypoalbuminemia	47	1	2	2	0	0	0
Hypertension	41	9	0	0	0	0	1
Proteinuria	40	7	2	0	4	8	6
Fatigue	38	1	2	2	0	0	0
Decrease in platelet count	34	3	1	1	0	0	0
Increase in ALT	28	1	0	0	0	0	0
Anemia	23	0	0	0	0	0	0
Pruritis	23	0	0	0	0	0	0
Anorexia	22	1	1	0	3	1	0
Bleeding or hemorrhage	19	4	2	0	1	1	0
Ascites	18	1	3	3	1	0	1
Fever	16	0	1	2	1	0	0
Decrease in white blood cells	15	5	0	0	0	0	0
Weight loss	14	0	0	0	0	0	0
Increase in blood bilirubin	12	0	0	3	1	0	0
Edema limbs	10	0	0	1	1	2	1
Diarrhea	9	0	0	0	0	0	0
Increase in creatinine	8	0	0	0	0	0	0
Decrease in neutrophil count	7	2	0	0	0	0	0
Nausea	5	1	0	0	0	0	0
Infection	5	4	4	0	0	1	0
Palmar-plantar erythrodysesthesia syndrome	4	0	0	0	0	0	0
Encephalopathy	3	3	0	2	0	0	0
Congestive heart failure	2	1	1	1	0	0	0
Thromboembolic event	2	1	0	2	0	0	0
Wound dehiscence	1	0	0	0	0	1	0
Invasive therapy associated with risk of wound healing complications during the administration of Bev							
Implantation of a CV access port	-	-	7	0	0	1	0
Operation for bone fracture	-	-	1	0	0	2	0
Therapies for the prevention of esophageal varix rupture	-	-	1	0	0	1	0

Adverse events were evaluated based on the Common Terminology Criteria for Adverse Events version 5.0. Variables are expressed as the total number. Abbreviations: AEs, adverse events; AST, aspartate aminotransferase; Atezo/Bev, atezolizumab plus bevacizumab therapy; ALT, alanine aminotransferase; Atezo, atezolizumab; Bev, bevacizumab; CV, central venous.

**Table 4 cancers-15-01568-t004:** Relationships between therapeutic modifications (*n* = 100).

Experience of Therapeutic Modification and AEs	Discontinuation of Both Atezo and Bev		Therapeutic Modifications without Discontinuation of Both Atezo and Bev (*n* = 46)	No Therapeutic Modification (*n* = 29)
	Without (*n* = 20)	With (*n* = 5)		
	Other Therapeutic Modifications			
Interruption of both Atezo and Bev (*n* = 35)	0 (0.0%)	3 (8.6%)	32 (91.4%)	0 (0.0%)
Discontinuation of both Atezo and Bev (*n* = 25)	20 (80.0%)	5 (20.0%)	0 (0.0%)	0 (0.0%)
Reduction in Bev dose (*n* = 12)	0 (0.0%)	0 (0.0%)	12 (100.0%)	0 (0.0%)
Interruption of Bev (*n* = 17)	0 (0.0%)	2 (11.8%)	15 (88.2%)	0 (0.0%)
Discontinuation of Bev (*n* = 8)	0 (0.0%)	1 (12.5%)	7 (87.5%)	0 (0.0%)
AEs of grade ≥3 (*n* = 31)	4 (12.9%)	4 (12.9%)	21 (67.7%)	2 (6.5%)
irAEs of any grade (*n* = 31)	11 (35.5%)	3 (9.7%)	11 (35.5%)	6 (19.4%)
AEs other than irAEs of grade ≥3 (*n* = 31)	3 (11.1%)	4 (14.8%)	18 (66.7%)	2 (7.4%)

Abbreviations: AEs, adverse events; Atezo, atezolizumab; Bev, bevacizumab; irAEs, immune-related adverse events.

**Table 5 cancers-15-01568-t005:** Cox regression analyses of factors contributing to (a) OS and (b) TTP, and logistic regression analyses of factors contributing to (c) the achievement of OR and (d) experience of discontinuation of both Atezo and Bev alone (*n* = 100).

	(a) OS				(b) TTP				(c) Achievement of OR				(d) Discontinuation of Both Atezo and Bev Alone			
Factors	Median OS (Months)	Uni	Multivariate		Median TTP (Months)	Uni	Multivariate		ORR (%)	Uni	Multivariate		DR (%)	Uni	Multivariate	
		*p*-Value	Hazard Ratio (95% CI)	*p*-Value		*p*-Value	Hazard Ratio (95% CI)	*p*-Value		*p*-Value	Odds Ratio (95% CI)	*p*-Value		*p*-Value	Odds Ratio (95% CI)	*p*-Value
Age																
<70 years	21.90	0.18			4.83	0.070			36.6	0.057			9.8	0.040	3.57 (1.00–11.21)	0.051
≥70 years	n.r.				8.19				55.9				27.1			
Sex																
Male	21.90	0.30			5.79	0.37			27.4	0.35			21.4	0.41		
Female	n.r.				5.00				18.8				12.5			
HBs antigen																
Positive	19. 2	0.12			n.r.	0.010		0.35	65.0	0.089			10.0	0.21		
Negative	21.90				5.46				43.8				22.5			
HCV antibody																
Positive	n.r.	0.52			5.16	0.44			50.0	0.90			26.7	0.28		
Negative	21.90				6.87				47.1				17.1			
Alcoholic liver disease																
Present	21.90	0.17			5.75	0.32			42.9	0.45			12.9	0.054		
Absent	19.20				5.89				50.8				31.3			
Esophageal varix																
Present	19.20	0.38			4.83	0.092			46.2	0.77			20.5	0.92		
Absent	n.r.				8.06				49.2				19.7			
Gastric varix																
Present	19.20	0.88			4.41	0.38			28.6	0.29			0.0	0.17		
Absent	21.90				5.89				49.5				20.5			
History of curative therapy																
Present	n.r.	0.43			6.87	0.80			44.4	0.44			18.5	0.69		
Absent	21.90				5.52				52.2				21.7			
History of TACE																
Present	21.90	0.76			5.75	0.35			44.6	0.36			20.0	1.00		
Absent	n.r.				7.73				54.3				20.0			
Child–Pugh class																
A	21.90	<0.001	reference	**0.0030**	7.33	0.0090		0.20	52.4	0.053			16.7	0.056		
B	7.30		4.24 (1.64–10.97)		2.30				25.0.				37.5			
mALBI grade																
1	n.r.				12.56				58.6				10.3		reference	
2a	21.90	0.71			5.75	0.25		0.38	51.9	0.64			14.8	0.43	n.a.	0.18
2b	n.r.	0.051			4.96	0.0020	.	0.99	37.2	0.061			30.2	0.026	n.a.	**0.025**
3	7.30	0.050			4.24	0.20		0.05	100.0	0.30			0.0	0.62	n.a.	0.79
ECOG-PS																
0	21.90	0.39			5.89	0.95			47.7	0.87			17.4	0.11		
1	n.r.				5.00				50.0				35.7			
Maximum tumor size																
≤6 cm	11.47	0.013		0.36	2.66	0.21			55.4	0.015	reference	**0.033**	23.1	0.65		
>6 cm	21.90				6.87				26.9		0.32 (0.11–0.91)		18.9			
Tumor number																
≤6	n.r.	0.19			8.61	0.11			56.6	0.067			14.9	0.23		
>6	21.90				5.46				38.3				24.5			
Intrahepatic tumor volume																
<50%	21.90	<0.001	9.72 (2.14–44.16)	**0.0030**	5.89	0.0010		0.26	49.5	0.091			19.6	0.56		
≥50%	5.39		reference		0.92				0.0				33.3			
Macrovascular invasion																
Present	19.20	0.14			4.41	0.34			20.0	0.027	0.24 (0.058–0.99)	**0.048**	26.7	0.48		
Absent	n.r.				6.87				52.9		reference		18.8			
Extrahepatic spread																
Present	n.r.	0.10			5.00	0.94			43.9	0.49			17.1	0.54		
Absent	21.90				6.81				50.8				22.0			
AFP concentration																
≥400 ng/mL	14.70	0.0090		0.19	3.78	0.022		0.092	28.6	0.018		0.20	19.4	0.82		
<400 ng/mL	21.90				7.33				55.6				21.4			
Number of chemotherapy lines																
First-line	21.90	0.31			6.81	0.18			60.3	0.0040	3.89 (1.59–9.53)	**0.0020**	22.4	0.48		
Second- or later-line	19.20				4.37				31.0		reference		16.7			
Best response																
Objective response	n.r.	<0.001	0.070 (0.015–0.32)	**<0.001**	10.00	<0.001	0.013 (0.0050–0.037)	**<0.001**					16.7			
Stable disease	15.35	0.0010	0.39 (0.14–1.09)	0.073	4.37	<0.001	0.050 (0.021–0.12)	**<0.001**					23.7			
Progressive disease	5.39		reference		0.92		reference						21.4			
Initial Bev dose																
Standard dose	11.24	0.31			5.89	0.30			48.4	0.71			24.2	0.43		
Reduced dose	21.90				3.49				40.0				40.0			
Experience of therapeutic modifications	During TTP				During TTP				Until TTBR							
Discontinuation of both Atezo and Bev alone	9.63	0.21	2.72 (1.05–7.07)	**0.040**	2.53	0.17	2.78 (1.40–5.51)	**0.0040**	36.4	0.61						
Discontinuation of both Atezo and Bev with other therapeutic modifications^†^	n.r.	0.15	0.41 (0.050–3.39)	0.41	n.r.	0.016	0.27 (0.057–1.28)	0.098	100.0	0.14						
Therapeutic modifications other than discontinuation of both Atezo and Bev^‡^	n.r.	<0.001	0.23 (0.075–0.68)	**0.0080**	10.0	0.0020	0.45 (0.24–0.87)	**0.016**	51.6	0.11						
No therapeutic modification	14.70		reference		4.24		reference		47.8							
IrAEs of any grade																
Experienced	21.90	0.60			8.61	0.093			58.8	0.37			35.5	0.012	3.57 (1.26–10.13)	**0.017**
Not experienced	n.r.				5.16				45.8				13.0		reference	
AEs other than irAEs																
Grade ≥ 3	n.r.	0.070			6.87	0.58			66.7	0.19			23.3	0.18		
Grade < 2	21.90				5.79				46.6				11.1			

Abbreviations: AEs, adverse events; AFP, alpha-fetoprotein; ALBI, albumin–bilirubin; Atezo, atezolizumab; Bev, bevacizumab; CI, confidence interval; DR, discontinuation rate; ECOG-PS, Eastern Cooperative Oncology Group performance status; HCV, hepatitis C virus; HBs, hepatis B surface; irAEs, immune-related adverse events; mOS, median overall survival; mTTP, median time to progression; n.a., not applicable; n.r., not reached; OR, objective response; ORR, objective response rate; OS, overall survival; TACE, transarterial chemoembolization; TTBR, time to confirmation of best response; TTP, time to progression; Uni, univariate. †Other therapeutic modifications include the interruption of Bev, discontinuation of Bev, and the interruption of both Atezo and Bev. ‡Other therapeutic modifications than the discontinuation of both Atezo and Bev include a reduction in Bev, interruption of Bev, discontinuation of Bev, and interruption of both Atezo and Bev. Bold font indicates significant *p* values.

**Table 6 cancers-15-01568-t006:** Relationships between best response and RDI during TTP, until TTBR, and after TTBR (*n* = 100).

RDI and Cycle	Timing	Best Response			*p*-Value
		OR (*n* = 48)	SD (*n* = 38)	PD (*n* = 14)	
RDI of Atezo	During TTP	0.88 (0.11–1.00)	0.96 (0.25–1.00)	1.00 (0.50–1.00)	0.069
	Until TTBR	1.00 (0.11–1.00)	1.00 (0.33–1.00)	1.00 (0.50–1.00)	0.66
	After TTBR	0.84 (0.00–1.00)	0.80 (0.00–1.00)	n.a.	0.062
RDI of Bev	During TTP	0.70 (0.11–1.00)	0.76 (0.18–1.00)	1.00 (0.50–1.00)	**0.016**
	Until TTBR	1.00 (0.11–1.00)	1.00 (0.30–1.00)	1.00 (0.50–1.00)	0.94
	After TTBR	0.69 (0.00–1.00)	0.80 (0.00–1.00)	n.a.	0.27
Number of administered cycles of Atezo	During TTP	9.0 (1–27)	5.0 (1–17)	1.5 (1–4)	**<** **0.001**
	Until TTBR	2.5 (1–13)	2.0 (1–3)	2.0 (1–4)	**<** **0.001**
	After TTBR	7.0 (0–26)	3.0 (0–16)	n.a.	**<** **0.001**
Number of administered cycles of Bev	During TTP	8.0 (1–27)	4.0 (1–17)	1.5 (1–3)	**<0.001**
	Until TTBR	2.0 (1–12)	2.0 (1–3)	2.0 (1–4)	**0.001**
	After TTBR	4.0 (0–18)	3.0 (0–14)	n.a.	**<0.001**

Abbreviations: Atezo, atezolizumab; Bev, bevacizumab; n.a., not applicable; OR, objective response; PD, progressive disease; RDI, relative dose intensity; SD, stable disease; TTBR, time to confirmation of best response; TTP, time to progression. Values are presented as the median (range). Bold font indicates statistically significant *p* values.

## Data Availability

Data that support the findings of this study are available from the author, T.T. (Takayuki Tokunaga), upon reasonable request.

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
