# Peer review of "Therapeutic Modifications without Discontinuation of Atezolizumab Plus Bevacizumab Therapy Are Associated with Favorable Overall Survival and Time to Progression in Patients with Unresectable Hepatocellular Carcinoma"

_cancers, 2023, doi:10.3390/cancers15051568_

Round 1

Reviewer 1 Report

Dear Authors

I would like to thank you for the opportunity of reviewing this interesting paper that is focused on a very remarkable and challenging topic that is a lively argument also in the daily clinical practice. 

The present study reveals the impact of therapeutic modifications in case of intolerable adverse events on the prognosis of patients with unresectable HCC receiving Atezo/Bev therapy. In particular, favorable prognosis was observed for patients in whom therapeutic modifications other than discontinuation of both Atezo and Bev were performed. In contrast, unfavorable prognosis was noted for patients in whom both Atezo and Bev were discontinued without other therapeutic modifications due to the occurrence of adverse events.

Therefore, papers that explore in depth this theme, that always represented a great challenge for all clinicians, especially in the era of tailored medicine, could surely be of interest for this important journal. Moreover, this paper demonstrates the aim of finding objective and practical conclusions from the many studies that have been conducted in recent years. 

This paper is pleasurable to read, although it suffers from some limitations that Authors can easily adjust in order to slightly improve their review making it more eligible for this important Journal. Furthermore, Authors can improve some section of the paper, adding information and including other important references about this topic that, in my opinion, should be cited and discussed. 

First of all, although language used is appropriate, I (I am not a native English speaker) recommend to Authors to obtain a certified native speaker with proficiencies in the scientific-medical field to complete properly this paper (if not jet done). Moreover, I recommend making a further revision of the manuscript to fix some small typing/language errors.

The title is clear and direct. Personally, I believe it could be improved and be more focused on the results, for example:“Therapeutic modifications without discontinuation of atezolizumab plus bevacizumab therapy are associated with favorable overall survival and time to progression in patients with unresectable hepatocellular carcinoma”. If the editor agrees, the authors could consider the change of the title.

Although the introduction fits the context of the study, it is concise. Sometimes, many concepts clearly explicated in an exhaustive introduction could help readers to become passionate about reading the paper and using it as a reference. For example, I suggest to add a few more sentences regarding the tumor immune microenvironment and how this plays a critical role in the pathogenesis of hepatocellular carcinoma [doi: 10.3389/fonc.2020.601240][doi: 10.3748/wjg.v25.i24.2977].  In fact, advances in understanding tumor immunity have led to a major shift in the treatment of HCC, with the emergence of immunotherapy where therapeutic agents are used to target immune cells rather than cancer cells. Moreover, some more information regarding the impact of immune-related adverse events caused by Atezo and Bev should be added. 

In the section “2.2. Radiological diagnosis of HCC”, please cite the following references [doi: 10.1016/j.jhep.2018.03.019][doi: 10.1002/hep.29913] [doi:10.1007/s11547-022-01449-w] and correct the sentence “stronger enhancement than that of the surrounding hepatic parenchyma in the arterial phase and weaker enhancement than that of the surrounding hepatic parenchyma in the portal venous or equilibrium phases” with “hypervascularity in late arterial phase and wash- out on portal venous and/or delayed phases”.

Moreover, in the section “2.3. Practice of Atezo/Bev”, I believe the last paragraph is redundant since the same concept is well stated in the following section 2.4. Authors should consider removing it.

Finally, In the section “2.4. Evaluation of Atezo/Bev”, regarding the sentence: “The best response to Atezo/Bev was evaluated according to modified Response 117 Evaluation Criteria in Solid Tumors (mRECIST) as complete response (CR), partial 118 response (PR), stable disease (SD), and progressive disease (PD)”. It is well known that the decision on whether a HCC patient is a responder or progressor under systemic therapy may vary among different operators, especially in case of a non-specifically trained radiologist and, therefore, regardless of the adopted criteria, patients should be evaluated by experienced radiologists to minimize variability in this critical instance [doi:10.1007/s00330-018-5393-3]. Please, could the Authors report the methodology used (an experienced radiologist evaluated CT and MRI?

Results remain clear and well-structured. However, in my opinion, they might result redundant and too long, since many of the data are already well presented in the many tables of the paper. The authors must decide to reduce the superfluous text which otherwise becomes difficult to read. 

Discussion is well presented.

Author Response

Point 1) First of all, although language used is appropriate, I (I am not a native English speaker) recommend to Authors to obtain a certified native speaker with proficiencies in the scientific-medical field to complete properly this paper (if not jet done). Moreover, I recommend making a further revision of the manuscript to fix some small typing/language errors.

Response 1) Thank you for indication. Our paper has been edited by a native speaker of English, who is familiar with scientific English. We carefully check the manuscript again and revised the errors. For one example, we deleted the duplications in Table 1.

Point 2) The title is clear and direct. Personally, I believe it could be improved and be more focused on the results, for example: “Therapeutic modifications without discontinuation of atezolizumab plus bevacizumab therapy are associated with favorable overall survival and time to progression in patients with unresectable hepatocellular carcinoma”. If the editor agrees, the authors could consider the change of the title.

Response 2) Thank you for advice. We changed the title into Therapeutic modifications without discontinuation of atezolizumab plus bevacizumab therapy are associated with favorable overall survival and time to progression in patients with unresectable hepatocellular carcinoma.

Point 3) Although the introduction fits the context of the study, it is concise. Sometimes, many concepts clearly explicated in an exhaustive introduction could help readers to become passionate about reading the paper and using it as a reference. For example, I suggest to add a few more sentences regarding the tumor immune microenvironment and how this plays a critical role in the pathogenesis of hepatocellular carcinoma. In fact, advances in understanding tumor immunity have led to a major shift in the treatment of HCC, with the emergence of immunotherapy where therapeutic agents are used to target immune cells rather than cancer cells. Moreover, some more information regarding the impact of immune-related adverse events caused by Atezo and Bev should be added. 

Response 3) Thank you for insightful suggestion. We cited [doi:10.3389/fonc.2020.601240] as a reference and added the sentences regarding tumor microenvironment of HCC and the impact of irAEs in Introduction section as follows:

[Introduction section]

Hepatocellular carcinoma (HCC) is classified into the inflamed class (35%) and the noninflamed class (65%) in the immunogenomic classification [1]. Although chemotherapy using single-agent immune checkpoint inhibitor (ICI) is considered not effective in the noninflamed HCC, a combination of ICI and anti-vascular endothelial growth factor (VEGF) antibody can be effective in such an immunosuppressive tumor microenvironment [2, 3].

In a phase 3 trial of first-line chemotherapy in patients with unresectable HCC (uHCC) (IMbrave150 trial), Atezolizumab (Atezo) plus Bevacizumab (Bev) therapy (Atezo/Bev) was associated with superior overall survival (OS), progression-free survival, and objective response (OR) rate compared with sorafenib therapy [4,5]. Atezo is an ICI that targets the programmed death ligand 1 (PD-L1) and blocks immune evasion by tumors. Bev is a monoclonal antibody that targets VEGF and blocks VEGF-mediated immunosuppression and angiogenesis in tumors. Immune-related adverse events (irAEs) caused by Atezo or adverse events (AEs) induced by Bev can occur during Atezo/Bev. In the IMbrave150 trial, interruption of Atezo or Bev and discontinuation of Atezo or Bev were carried out in response to the occurrence of intolerable AEs [4,5]. In a phase 2 study of Bev monotherapy in patients with uHCC, reduction of the Bev dosage was also permitted [6]. However, the impacts of therapeutic modifications and AEs, including irAEs, on the outcome of Atezo/Bev remain to be clarified.

In this study, the impact of therapeutic modifications on the OS, time to progression (TTP), and achievement of OR in patients with uHCC receiving Atezo/Bev was assessed. The aim was to determine the optimal management in real-world practice.

Point 4) In the section “2.2. Radiological diagnosis of HCC”, please cite the following references. and correct the sentence “stronger enhancement than that of the surrounding hepatic parenchyma in the arterial phase and weaker enhancement than that of the surrounding hepatic parenchyma in the portal venous or equilibrium phases” with “hypervascularity in late arterial phase and wash- out on portal venous and/or delayed phases”.

Response 4) Thank you for helpful indication. We cited [doi:10.1016/j.jhep.2018.03.019] [doi:10.1002/hep.29913] [doi:10.1007/s11547-022-01449-w] as references, and revised the sentence of Method section (2.2) as follows: hypervascularity in late arterial phase and wash-out on portal venous and/or delayed phases.

Point 5) Moreover, in the section “2.3. Practice of Atezo/Bev”, I believe the last paragraph is redundant since the same concept is well stated in the following section 2.4. Authors should consider removing it.

Response 5) Thank you for helpful indication. We partially removed the paragraph and revised the remaining sentence [Methods section 2.3.].

Point 6) Finally, In the section “2.4. Evaluation of Atezo/Bev”, regarding the sentence: “The best response to Atezo/Bev was evaluated according to modified Response Evaluation Criteria in Solid Tumors (mRECIST) as complete response (CR), partial 118 response (PR), stable disease (SD), and progressive disease (PD)”. It is well known that the decision on whether a HCC patient is a responder or progressor under systemic therapy may vary among different operators, especially in case of a non-specifically trained radiologist and, therefore, regardless of the adopted criteria, patients should be evaluated by experienced radiologists to minimize variability in this critical instance [doi:10.1007/s00330-018-5393-3]. Please, could the Authors report the methodology used (an experienced radiologist evaluated CT and MRI?

Response 6) Thank you for insightful comment. The radiological Evaluation of Atezo/Bev in this study was conducted by experienced radiologists and hepatologists. We revised the manuscript as follows: The best response to Atezo/Bev was evaluated by the experienced radiologists and hepatologists according to modified Response Evaluation Criteria in Solid Tumors (mRECIST)[Methods section 2.4.].

Point 7) Results remain clear and well-structured. However, in my opinion, they might result redundant and too long, since many of the data are already well presented in the many tables of the paper. The authors must decide to reduce the superfluous text which otherwise becomes difficult to read. 

Response 7) Thank you for helpful indication. We reduced the sentenced in the Result section as far as possible [Results section 3.1 and 3.3].

Reviewer 2 Report

The Authors dealt with a very interesting topic. Overall, the paper is well-written and timely-needed, but there are a number of critical methodological issues deserving attention.

1) Abstract: "Therapeutic modifications without discontinuation of both Atezo and Bev (n=46) were associated with favourable overall survival (median not reached; hazard ratio [HR]:0.23)". It is not clear which is the reference for the hazard ratio: discontinuation followed by other therapies. Other?

1) Inclusion and exclusion criteria (line 80). Oesophagal varices were not mentioned, but their role in the prescrivibility of bevacizumab and their management during treatment are hot topics. How did the authors deal with patients with high-risk varices at the screening? Were they included? Did they receive band ligation and/or non-selective beta blockers before starting treatment?

2) Study design (line 111): " Progression was defined as radiological progression, or discontinuation of both Atezo and Bev because of intolerable AEs despite management". This definition is odd. Progression is progression (radiological or clinical), intolerance is quite a different thing. Please revise.

3) Results (line 189): "The median OS was 21.90 months (95% confidence interval [CI]: 16.86–26.93 months), and the median TTP was 5.79 months (95% CI: 3.69–7.88 months)". It is quite a long post-progression survival, hinting at subsequent therapies. Please report any subsequent locoregional or systemic treatment. Also, consider adding further treatments as a variable in the multivariable models of OS. 

4) Results (Table 1): This table can be simplified. There are a number of itemswhich do not require two different rows. For instance, intrahepatic tumour volume should not be reported as yes/no, but using a single "intrahepatic tumour volume>50%" row (without the absent/present dichotomy). Also, there are some duplicated items in this table.

5) Results: This study has a considerable inhomogeneity in therapeutic lines. Multivariable models of OS, OR, and TTP should contain a frontline vs subsequent line variable, as immunotherapy could be less effective in TKI-pretreated patients (Haber et al, Gastoenterology 2022).

6) Results: the most critical point of this study is the lack of time-dependent analyses. This flaw can lead to distorted results. For instance, in Table 5, the authors found that objective response was a strong predictor of survival without considering that an immortal-time bias can be found as patients who died before performing the follow-up CT scan due to tumour progression are excluded. To a similar, if not greater extent, patients experiencing a dose reduction/interruption might seem to live longer for the simple fact that early progressors had a shorter time to develop adverse events before dying. The authors seem to know the time between the start of atezo-bev and the therapeutic modification (or to the first imaging follow-up), so this kind of analysis should not be complex.

Author Response

Point 1) Abstract: "Therapeutic modifications without discontinuation of both Atezo and Bev (n=46) were associated with favorable overall survival (median not reached; hazard ratio [HR]:0.23)". It is not clear which is the reference for the hazard ratio: discontinuation followed by other therapies. Other?

Response 1) Thank you for your comment about the lack of clarity about the reference. We revised the Abstract section as follows:

[Abstract] “Therapeutic modifications without discontinuation of both Atezo and Bev (n=46) were associated with favorable overall survival (median not reached; hazard ratio [HR]:0.23) and time to progression (median:10.00 months; HR:0.23) with no therapeutic modification defined as reference.

Point 2) Inclusion and exclusion criteria (line 80). Oesophagal varices were not mentioned, but their role in the prescrivibility of bevacizumab and their management during treatment are hot topics. How did the authors deal with patients with high-risk varices at the screening? Were they included? Did they receive band ligation and/or non-selective beta blockers before starting treatment?

Resonse 2) Thank you for the important indication. All the patients in this study were assessed with esophagogastroduodenoscopy before the initiation of Atezo/Bev. The initiation of Atezo/Bev preceded the treatment of varix with high risk of bleeding, in the case progression of HCC was predicted during the treatment of varix. Thirty-nine patients had esophageal varices. Seven of them had the varices with high risk of bleeding, but they started Atezo/Bev without receiving preventative endoscopic therapy because progression of HCC was predicted during treatment of the varices. One patient, who had high-risk esophageal varix at the screening, experienced the rupture, resulting in interruption of both Atezo and Bev. Seven patients had gastric varices, all of which were low risk of bleeding, and no patients experienced the rupture.

We revised the Method and Result section as follows:

[Methods section 2.1.] “All the patients in this study were assessed with esophagogastroduodenoscopy before the initiation of Atezo/Bev. The initiation of Atezo/Bev preceded the treatment of high-risk varix, in the case progression of HCC was predicted during the treatment of varix.

We also added the Result section as follows:

 [Result section 3.1.] “Thirty-nine patients had esophageal varices. Seven of them had the varices with high risk of bleeding, but they started Atezo/Bev without receiving preventative endoscopic therapy because progression of HCC was predicted during treatment of the varices. Seven patients had gastric varices, all of which were low risk of bleeding.

[Result section 3.3.] “One patient, who had high-risk esophageal varix at the screening, experienced the rupture, resulting in interruption of both Atezo and Bev.

Furthermore, we added esophageal varix and gastric varix into the analyzed factor of OS, TTP, achievement of OR and discontinuation of both Atezo and Bev without other therapeutic modifications (Table 5). Although neither esophageal varix and gastric varix were associated with OS, TTP, achievement of OR and discontinuation of both Atezo and Bev without other therapeutic modifications in this study, receiving preventative endoscopic therapy before the initiation of Atezo/Bev would be desirable in line with the IMbrave 150 trial, in the case high-risk varix was detected at the screening and the patients had time to spare. The management of high-risk varix in the patients who had no time to spare to prevent the bleeding before Atezo/Bev should be assessed in the future.

We added the Discussion section as follows:

[line 454-460] “Although neither esophageal varix and gastric varix were associated with OS, TTP, achievement of OR and discontinuation of both Atezo and Bev without other therapeutic modifications in this study, receiving preventative endoscopic therapy before the initiation of Atezo/Bev would be desirable in line with the IMbrave 150 trial, in the case high-risk varix was detected at the screening and the patients had time to spare. The management of high-risk varix in the patients who had no time to spare to prevent the bleeding before Atezo/Bev should be assessed in the future.

Point 3) Study design (line 111): " Progression was defined as radiological progression, or discontinuation of both Atezo and Bev because of intolerable AEs despite management". This definition is odd. Progression is progression (radiological or clinical), intolerance is quite a different thing. Please revise.

Response 3) Thank you for your comment about the lack of clarity about the definition of progression. Actually, we defined progression as not only radiological progression but also clinical progression such as deterioration of liver function or performance status. Sixty-six patients experienced progression. Among them, 58 patients had radiological progression without clinical progression, 2 patients had both radiological and clinical progression, and 6 patients had clinical progression without radiological progression.

We revised the sentence in Method section as follows, [line116-117] Atezo/Bev was continued until radiological or clinical progression”.

We added Methods section as follows, [line200-202] Sixty-six patients experienced progression. Among them, 58 patients had radiological progression without clinical progression, 2 patients had both radiological and clinical progression, and 6 patients had clinical progression without radiological progression.

Point 4) Results (line 189): "The median OS was 21.90 months (95% confidence interval [CI]: 16.86–26.93 months), and the median TTP was 5.79 months (95% CI: 3.69–7.88 months)". It is quite a long post-progression survival, hinting at subsequent therapies. Please report any subsequent locoregional or systemic treatment. Also, consider adding further treatments as a variable in the multivariable models of OS.”

Response 4) Thank you for insightful suggestions. Thirty-one patients received subsequent therapy, including cabozantinib therapy (n=1), continuation of atezolizumab and bevacizumab therapy beyond progression (n=16), hepatic arterial infusion chemotherapy (n=4), lenvatinib therapy (n=2), radiation therapy (n=1), ramucirumab therapy (n=2) and TACE (n=5).We analyzed factors contributing to OS, including subsequent therapy as a variable, in the patients who progressed on Atezo/Bev, and we found therapeutic modifications other than discontinuation of both atezolizumab and bevacizumab was a favorable factor with no therapeutic modification defined as reference (Table S1). We also found that practice of subsequent therapy was not independently associated with OS. Therefore, it is conceivable that therapeutic modifications other than discontinuation of both atezolizumab and bevacizumab would be positively associated with OS.

We revised the Result section as follows,

[Result section 3.4.]In the patients who progressed on Atezo/Bev, achievement of OR (HR: 0.11; 95% CI: 0.024–0.50; P=0.0040) and experience of therapeutic modifications other than discontinuation of both Atezo and Bev (HR: 0.24; 95% CI: 0.079–0.70; P=0.0090) were favorable factors of OS with PD and no therapeutic modification used as reference, respectively, while experience of subsequent therapy (n=31) was not an independent factor of OS (Table S1).

Point 5) Results (Table 1): This table can be simplified. There are a number of items which do not require two different rows. For instance, intrahepatic tumour volume should not be reported as yes/no, but using a single "intrahepatic tumour volume>50%" row (without the absent/present dichotomy). Also, there are some duplicated items in this table.

Response 5) Thank you for your indication. We deleted the duplications. Furthermore, we changed two different rows to a single row as far as possible.

Point 6) Results: This study has a considerable inhomogeneity in therapeutic lines. Multivariable models of OS, OR, and TTP should contain a frontline vs subsequent line variable, as immunotherapy could be less effective in TKI-pretreated patients (Haber et al, Gastoenterology 2022).

Response 6) Thank you for your question. We already have shown the association between the number of chemotherapy line and OS, TTP and OR (First-line/Second- or later- line) in the first proof. We found that the number of chemotherapy line were not associated with OS and TTP, while practice of atezolizumab and bevacizumab therapy as first-line positively contributed to achievement of OR (Table 5). Furthermore, we have already stated these findings in Results and Discussion sections [line 188, 348-349, 448-453].

Point 7) Results: the most critical point of this study is the lack of time-dependent analyses. This flaw can lead to distorted results. For instance, in Table 5, the authors found that objective response was a strong predictor of survival without considering that an immortal-time bias can be found as patients who died before performing the follow-up CT scan due to tumour progression are excluded. To a similar, if not greater extent, patients experiencing a dose reduction/interruption might seem to live longer for the simple fact that early progressors had a shorter time to develop adverse events before dying. The authors seem to know the time between the start of atezo-bev and the therapeutic modification (or to the first imaging follow-up), so this kind of analysis should not be complex.

Response 7) Thank you for insightful comment about the immortal-time bias in this study. We additionally performed landmark analyses of OS and TTP regarding best response (Figure S1 and S2) and therapeutic modifications (Figure S3 and S4), using landmark time at 2, 4 and 6 months. Furthermore, we analyzed factors associated with OS and TTP in the patients with OR and SD, excluding the patients with PD (Table S2).

Regarding landmark analysis of best response, we found that:

ï½¥ the patients with OR had favorable OS compared with SD at 2, 4 and 6 months (p=0.039, 0.001 and <0.001) and PD at 2, 4 and 6 months (p<0.001, <0.001 and <0.001) (Figure S1).

ï½¥ The patients with OR had favorable TTP compared with PD at 2 months (p<0.001), while those with OR had similar TTP compared with SD at 2, 4 and 6 months (Figure S2).

Concerning landmark analysis of therapeutic modifications, we found that:

ï½¥ the patients with therapeutic modifications other than discontinuation of both Atezo and Bev had favorable OS at 2 months (p<0.001), similar OS at 4 and 6 months, and similar TTP at 2, 4 and 6 months, compared with those with no therapeutic modification (Figure S3, S4).

ï½¥ The patients with discontinuation of both Atezo and Bev alone had unfavorable OS at 2 months (p<0.001), and unfavorable TTP at 2 and 4 months (p=0.014 and 0.02) (Figure S3, S4).

In the patients who progressed on Atezo/Bev, achievement of OR (HR: 0.11; 95% CI: 0.024–0.50; P=0.0040) and experience of therapeutic modifications other than discontinuation of both Atezo and Bev (HR: 0.24; 95% CI: 0.079–0.70; P=0.0090) were also favorable factors of OS with PD and no therapeutic modification used as reference, respectively, while experience of subsequent therapy (n=31) was not an independent factor of OS (Table S1).

Furthermore, in the patients with OR and SD, achievement of OR (HR: 0.085; 95% CI: 0.018–0.40; P=0.0020) was also associated with favorable OS (HR: 4.93; 95% CI: 1.37–17.74; P=0.015) and favorable TTP (HR: 0.26; 95% CI: 0.14–0.47; P<0.001), while experience of discontinuation of both Atezo and Bev was also associated with unfavorable OS (HR: 4.93; 95% CI: 1.37–17.74; P=0.015) and unfavorable TTP (HR: 3.59; 95% CI: 1.57–8.23; P=0.0030), with no therapeutic modification used as reference, while therapeutic modifications other than discontinuation of both Atezo and Bev as well as discontinuation of both Atezo and Bev with other therapeutic modifications were not associated with OS or TTP (Table S2).

We hope that these results of additional analyses meet with your approval of the impact of therapeutic modifications of Atezo/Bevas. We changed the title into Therapeutic modifications without discontinuation of atezolizumab plus bevacizumab therapy are associated with favorable overall survival and time to progression in patients with unresectable hepatocellular carcinoma, following the reviewer 1’s advice. We revised Method, Result sections as follows:

[Method section, 2.6. Study assessment]

Firstly, factors associated with OS and TTP, including characteristics at the initiation of Atezo/Bev, best response, experience of AEs during TTP, and experience of therapeutic modifications during TTP, were analyzed in all the patients as well as in the patients with OR and SD at best response. Regarding time-dependent factors of best response and experience of therapeutic modifications, landmark analyses of OS and TTP at 2, 4, and 6 months after the initiation were performed. Patients who died and progressed before the landmark point were excluded from analysis of OS and TTP, respectively.

[Result section, 3.4. Factors associated with OS and factors associated with TTP]

In all the enrolled patients, achievement of OR was a favorable factor of OS (hazard ratio [HR]: 0.070; 95% CI: 0.015–0.32; P<0.001) and TTP (HR: 0.013; 95% CI: 0.0050–0.037; P<0.001), with PD used as reference. Experience of therapeutic modifications other than discontinuation of both Atezo and Bev was also associated with favorable OS (HR: 0.23; 95% CI: 0.075–0.68; P=0.0080) and favorable TTP (HR: 0.27; 95% CI: 0.057–1.28; P=0.0040), with no therapeutic modification used as reference. Nevertheless, experience of discontinuation of both Atezo and Bev alone was associated with unfavorable OS (HR: 2.72; 95% CI: 1.05–7.07; P=0.040) and unfavorable TTP (HR: 2.72; 95% CI: 1.05–7.07; P=0.040), with no therapeutic modification used as reference. Notably, discontinuation of both Atezo and Bev with other therapeutic modifications was not associated with OS or TTP (Table 5).

     In the patients who progressed on Atezo/Bev, achievement of OR (HR: 0.11; 95% CI: 0.024–0.50; P=0.0040) and experience of therapeutic modifications other than discontinuation of both Atezo and Bev (HR: 0.24; 95% CI: 0.079–0.70; P=0.0090) were also favorable factors of OS with PD and no therapeutic modification used as reference, respectively, while experience of subsequent therapy (n=31) was not an independent factor of OS (Table S1).

     Furthermore, in the patients with OR and SD, achievement of OR (HR: 0.085; 95% CI: 0.018–0.40; P=0.0020) was also associated with favorable OS (HR: 4.93; 95% CI: 1.37–17.74; P=0.015) and favorable TTP (HR: 0.26; 95% CI: 0.14–0.47; P<0.001), while experience of discontinuation of both Atezo and Bev was also associated with unfavorable OS (HR: 4.93; 95% CI: 1.37–17.74; P=0.015) and unfavorable TTP (HR: 3.59; 95% CI: 1.57–8.23; P=0.0030), with no therapeutic modification used as reference, while therapeutic modifications other than discontinuation of both Atezo and Bev as well as discontinuation of both Atezo and Bev with other therapeutic modifications were not associated with OS or TTP (Table S2).

[Result section, 3.5. Land mark analyses of OS and TTP regarding best response and therapeutic modifications]

     Regarding best response, the patients with OR had favorable OS compared with SD at 2, 4 and 6 months (p=0.039, 0.001 and <0.001) or PD at 2, 4 and 6 months (p<0.001, <0.001 and <0.001) (Figure S1). The patients with OR had favorable TTP compared with PD at 2 months (p<0.001), while those with OR had similar TTP compared with SD at 2, 4 and 6 months (Figure S2).

     Concerning therapeutic modifications, the patients with therapeutic modifications other than discontinuation of both Atezo and Bev had favorable OS at 2 months (p<0.001), similar OS at 4 and 6 months, and similar TTP at 2, 4 and 6 months, compared with those with no therapeutic modification (Figure S3, S4). The patients with discontinuation of both Atezo and Bev alone had unfavorable OS at 2 months (p<0.001), and unfavorable TTP at 2 and 4 months (p=0.014 and 0.02) (Figure S3, S4).

Round 2

Reviewer 1 Report

Authors addressed raised points appropriately.

Reviewer 2 Report

I would like to thank the authors for considering all the raised points. While time-dependent Cox regression would have been the best tool to address time-dependent variables, landmark analyses are an acceptable and widely used alternative in HCC studies. Therefore, I have no additional comments and suggest accepting this manuscript.